# Ballistic transport and boundary resistances in inhomogeneous quantum spin chains

Alberto Biella[1]*, Mario Collura [2,3,4], Davide Rossini [5], Andrea De Luca[6,7] & Leonardo Mazza[8]

Transport phenomena are central to physics, and transport in the many-body and fully-quantum regime is attracting an increasing amount of attention. It has been recently revealed that some quantum spin chains support ballistic transport of excitations at all energies. However, when joining two semi-infinite ballistic parts, such as the XX and XXZ spin-1/2 models, our understanding suddenly becomes less established. Employing a matrix-product-state ansatz of the wavefunction, we study the relaxation dynamics in this latter case. Here we show that it takes place inside a light cone, within which two qualitatively different regions coexist: an inner one with a strong tendency towards thermalization, and an outer one supporting ballistic transport. We comment on the possibility that even at infinite time the system supports stationary currents and displays a non-zero Kapitza boundary resistance. Our study paves the way to the analysis of the interplay between transport, integrability, and local defects.

[1] Université de Paris, Laboratoire Matériaux et Phénomènes Quantiques, CNRS, F-75013 Paris, France. [2] Theoretische Physik, Universität des Saarlandes, D-66123 Saarbrücken, Germany. [3] Dipartimento di Fisica e Astronomia "G. Galilei", Università di Padova, I-35131 Padova, Italy. [4] SISSA – International School for Advanced Studies, I-34136 Trieste, Italy. [5] Dipartimento di Fisica, Università di Pisa and INFN, Largo Pontecorvo 3, I-56127 Pisa, Italy. [6] The Rudolf Peierls Centre for Theoretical Physics, Oxford University, Oxford OX1 3NP, UK. [7] Laboratoire de Physique Théorique et Modélisation (CNRS UMR 8089), Université de Cergy-Pontoise, F-95302 Cergy-Pontoise, France. [8] LPTMS, UMR 8626, CNRS, Université Paris-Sud, Université Paris-Saclay, 91405 Orsay, France. *email: alberto.biella@univ-paris-diderot.fr

In 1941, P. Kapitza reported on the first measurement of the temperature drop near the boundary between helium and a solid when heat flows across the boundary[1]. The phenomenon, unrelated to that of conventional contact resistances, is ascribed to the mismatch between the energy carriers of the two materials, and appears even if the interface is perfect at the atomic scale. In the years 1951–1956, Kalatchnikov first, and Mazo and Onsager later, independently developed the so-called acoustic mismatch model, that gives a mathematical formulation to such intuition[2,3]. The quantitative comparison between experimental data and theoretical predictions motivated extensive investigations even in other interfaces without helium, e.g., between two solids[2], where the phenomenon has been often recovered.

Owing to the difficulty of performing numerical simulations in order to benchmark the theory, simpler treatable models have been considered. In particular, focusing on junctions of classical one-dimensional (1D) harmonic chains enabled to quantitatively investigate the phenomenon of *thermal boundary resistances*[4–6]. The effect has been widely reproduced, and comparisons with suitable adaptations of the theory have been presented. Yet, a numerical investigation of the phenomenon in the fully quantum case has not been performed so far[7].

The recent advances in the analytical understanding of the non-equilibrium dynamics of quantum spin chains[8–12] stimulated the improvement of state-of-the-art numerical simulations addressing the unitary time-evolution of strongly-correlated 1D quantum models[13–15]. Groundbreaking cold-atom experiments have also investigated several aspects of the coherent dynamics of closed quantum many-body systems[16–24]. These research efforts allowed to faithfully address how the presence of impurities and inhomogeneities may affect the quantum transport phenomena, also far from the linear-response regime[25–29].

As a paradigmatic example, we study a setup composed of two different semi-infinite spin-1/2 chains connected through a junction at $x = 0$ (see Fig. 1)[27]. In order to highlight the resistive effects taking place at the junction, we consider two models for which ballistic transport of spin and energy has been unambiguously demonstrated. For $x<0$, the spin chain can be mapped to non-interacting fermions: it supports ballistic transport, because the back-scattering phenomena due to interactions are impossible[30,31]. For $x>0$, we take a class of models that are solvable through Bethe ansatz (and thus integrable), where transport phenomena can be modelled by means of a generalized hydrodynamic theory (GHD)[32,33]. The latter predicts ballistic transport, notwithstanding the presence of particle-particle interactions, due to elastic scattering without backscattering[34]. While the dynamics of both halves can be described in terms of stable quasiparticles, the microscopic nature of such quasiparticles is different and therefore a nontrivial scattering dynamics is expected at the junction.

Here we present a detailed numerical study of the long-time behaviour of this setup, by considering the following protocol: (i) we prepare the system in a pure state where the two halves $x0$ have an extensive imbalance of a global conserved quantity; (ii) we unitarily evolve the state and monitor how such an imbalance is redistributed in time. In particular, we focus on two very representative examples, where at the beginning either the magnetization or the energy are macroscopically different within the two halves. For high temperatures, density matrices proved to be effective in exploring the linear response regime at long times[15]. Here instead, we focus on large imbalances, where pure states are numerically more tractable. Moreover, this approach allows us to naturally access the real-time entanglement dynamics. We demonstrate that the relaxation dynamics takes place inside a light cone, within which two regions can be distinguished: the inner one, close to the junction, displays a strong tendency

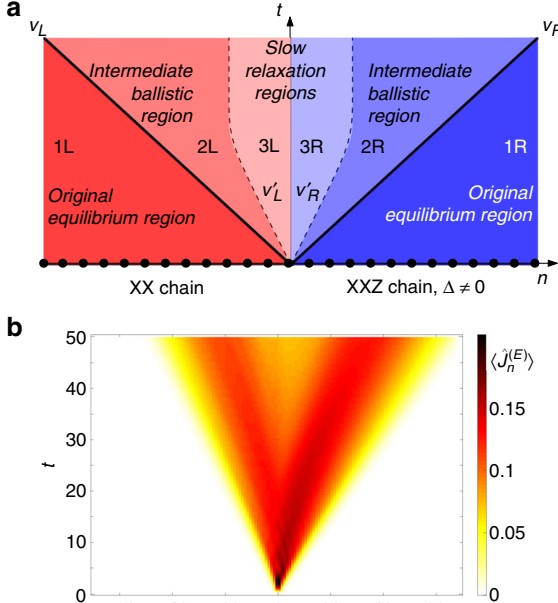

**Fig. 1** Transport in inhomogeneous spin chains. **a** An inhomogeneous quantum spin-1/2 chain composed of an XX and of an XXZ half is initialized in a pure state with energy or magnetization imbalance and subsequently evolved in time. The relaxation dynamics inside the light cone (thick black lines, velocities $v_{L,R}$) witnesses the coexistence of two (intermediate) ballistic regions and of two slow relaxation regions. The slow dynamics initially takes place inside an internal light cone with velocities $v'_{L,R}$, while, at long times, the extension of these regions remains constant in time. **b** Space-time profile of the energy current for $\Delta = \cos(\pi/4)$ — see Eq. (1). At long enough times, a decay of the current at the junction appears as a consequence of a thermalization process

towards relaxation; the outer one supports ballistic transport. We discuss how the system could evolve at times longer than those accessible with our numerical methods and argue that a Kapitza boundary resistance could appear.

## Results

**Overview**. The emerging scenario is sketched in the top panel of Fig. 1 and summarized below. As already observed in other set-ups, two wavefronts propagating at different velocities $v_L$ and $v_R$ emerge from the junction[27,33,35,36]; $v_L$ and $v_R$ correspond to the velocities of the fastest quasiparticle excitations of the left and right halves, respectively. Outside this light-cone, in agreement with the Lieb-Robinson bound[37], regions 1L and 1R display the initial equilibrium behaviour and are unaffected by the dynamics: these are the original equilibrium regions. Conversely, within the causal region, the system properties are nontrivially affected by the unitary dynamics.

As a first key result, we find that, when two different ballistic models are joined together, the relaxation dynamics inside the light-cone exhibits two qualitatively different behaviours and time scales, depending on whether we are observing the system near or far from the junction. Close to the edges of the light-cone (regions 2L and 2R in Fig. 1), a stationary state supporting a stable current flow is rapidly approached; these regions are accordingly dubbed 'intermediate ballistic regions'. Otherwise, around the junction, the current intensity keeps decreasing without reaching any stationary value even at the largest accessible time in our numerics. As a matter of fact, regions 3L and 3R are characterized by a slow relaxation dynamics and are named 'slow relaxation

regions'. As an example of the data supporting our conclusions, in the bottom panel of Fig. 1 we provide the space-time profile of the energy current for a paradigmatic case (see Results for further details).

We can interpret our results using a semiclassical picture of quasiparticles that travel freely inside each half, but undergo scattering events with frequency $\sim 1/\tau_{L,R}$ in a region $[-\ell_L, \ell_R]$ around the junction. Fast quasiparticles spend less time around the junction and are therefore less affected by these processes; this creates two qualitative space-time regions, separated by interfaces at $x/t \simeq v'_L \simeq \ell_L/\tau_L$ and $x/t \simeq v'_R \simeq \ell_R/\tau_R$. Note that $|v'_{L,R}| < |v_{L,R}|$. The regions 2L and 2R are forming because of the fast quasiparticles which pass almost unchanged through the junction: this justifies a quick relaxation to a stationary state on each ray at constant $x/t$, similarly to the homogeneous case. On the other hand, regions 3L and 3R are characterized by several scattering events, and display a strong tendency towards equilibration.

As a second crucial result, we propose an extrapolation of the data at long time. We argue that the standard scenario with thermalization in the whole system, eventually leading to vanishing energy and spin currents everywhere, is incompatible with the ballistic transport in the bulk of each half. At long times, the system exhibits a local quasi-stationary state (LQSS) on each ray $x/t$, as it happens for homogeneous integrable ballistic systems, but with a non-trivial behaviour in a region around the junction, whose width does not grow with time. Within this framework, the two emerging velocities $v'_{L,R}$ must tend to zero at long times (i.e. for $t \gg |\ell_{L,R}/v'_{L,R}|$), as depicted in the top panel of Fig. 1.

Some peculiar aspects of this emerging asymptotic scenario are retrieved in a simplified Ohmic model where the two semi-infinite ballistic spin chains are coupled via a diffusive junction of fixed width $\ell$, which is much larger than the typical relaxation length. The long-time dynamics of such a model can be solved within the hydrodynamic framework that describes ballistic transport in each half, and the resulting stationary profiles exhibit certain distinguishing features of the exact numerical simulations. Among them, the most important one is the presence of a sharp discontinuity in several local quantities as $x/t \to 0^{\pm}$, together with a long lived energy/spin current. This phenomenology is analogous to that of the thermal Kapitza boundary resistances, which is witnessed here in a fully-quantum scenario without important approximations.

**Setup**. We consider a quantum spin-1/2 chain described by the Hamiltonian $\hat{H} = \sum_n \hat{h}_n$:

$$\hat{h}_n = J(\hat{S}_n^x \hat{S}_{n+1}^x + \hat{S}_n^y \hat{S}_{n+1}^y + \Delta_n \hat{S}_n^z \hat{S}_{n+1}^z), \quad J > 0, \quad (1)$$

where $\hat{S}_n^\alpha$ is the operator associated to the $\alpha$ component of the $n$-th spin (hereafter we adopt units of $\hbar = k_B = 1$). The anisotropy parameter is space-dependent: for $n \leq 0$, $\Delta_n = 0$ whereas for $n > 0$, $\Delta_n = \Delta$. This model describes a perfect junction between a XX model and a XXZ model; a Jordan-Wigner transformation maps the former into non-interacting spinless fermions and the latter into interacting ones. We fix $\Delta = \cos(\gamma) \in [0, 1)$ so as to have a gapless model whose transport properties are well understood in the uniform regime.

At $t = 0$, the system is initialized in a pure state. We consider two kinds of initial states. The first one is the ground state $|\Psi_1\rangle$ of the Hamiltonian $\hat{H}' = \sum_n \lambda_n \hat{h}_n$, where $\lambda_n = -1$ for $n \leq 0$ and $\lambda_n = 1$ for $n > 0$. In this way, for $n \leq 0$ the state locally approximates the maximally excited state of the XX chain, whereas for $n > 0$ it approximates the ground state of the XXZ

chain. The state $|\Psi_1\rangle$ is thus the starting point of a partitioning protocol with two different inverse temperatures, $\beta_L$ and $\beta_R$, in the limit $\beta_{L,R} \to \mp\infty$. In this way, the energy imbalance is maximal and the state is pure. The second one is the domain-wall state[36,39,40] $|\Psi_2\rangle = \otimes_n |\psi_n\rangle$, with $|\psi_n\rangle = |\uparrow\rangle$ for $n \leq 0$ and $|\psi_n\rangle = |\downarrow\rangle$ for $n > 0$, which corresponds to maximal magnetization imbalance.

Thereafter, the system evolves unitarily as $|\Psi_{1,2}(t)\rangle = e^{-i\hat{H}t} |\Psi_{1,2}\rangle$, and allows the use of algorithms based on matrix-product-states for pure states (see "Methods"); previous studies with thermal states were not able to access novel equilibration regimes[27]. We focus on two different sets of observables: in the first case, we look at the energy transport, through the local energy density, $E_n = \langle \hat{h}_n \rangle$, and energy current, $\hat{J}_n^{(E)} = \frac{i}{2}([\hat{h}_n, \hat{h}_{n+1}] - [\hat{h}_n, \hat{h}_{n-1}])$. In the second one, we focus on the spin transport and measure the local magnetization, $S_n^z = \langle \hat{S}_n^z \rangle$, and the associated current $\hat{J}_n^{(M)} = iJ(\hat{S}_n^y \hat{S}_{n+1}^y - \hat{S}_{n+1}^x \hat{S}_n^y)$. Throughout this article, time is measured in units of $J^{-1}$, energy in units of $J$, energy current in units of $J^2$ and magnetization current in units of $J$.

Since the dynamics takes place inside a light cone (see Fig. 1), we will mostly present our numerical data using rescaled space-time units $x/t$. As an aside, this also permits to translate the problem in the standard framework of transport between two reservoirs held at fixed distance[10,41].

**Numerical results for the real-time dynamics**. We begin with the study of energy transport in the first situation, where the system is initially prepared in $|\Psi_1\rangle$. In Fig. 2a, we plot the local energy density profile $E_n(t)$ for $\gamma = \pi/4$. We observe that, for the rays close to the junction $|n|/t \lesssim 0.5$, energy is redistributed in the form of two approximately flat plateaus. In Fig. 2b, we plot the energy profile at the largest accessible time, $t = 40$, for several values of $\gamma$: the plateaus appear whenever $\Delta \neq 0$ and their width increases monotonically with $\Delta > 0$.

A similar study can be performed for the domain-wall initial state $|\Psi_2\rangle$, when focusing on the spin transport; as a matter of fact, the results are analogous to those obtained above for the

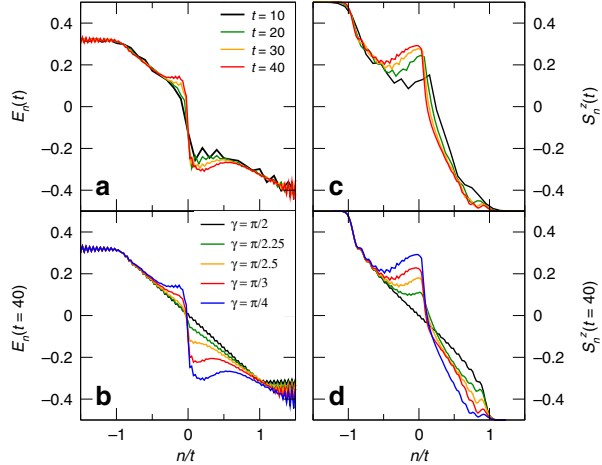

**Fig. 2** Energy and magnetization profiles during the relaxation process. Energy-transport scenario: **a** energy profile for $\gamma = \pi/4$ and several values of time; **b** energy profile at the longest accessible time, $t = 40$, and several values of $\gamma$. Spin-transport scenario: **c** magnetization profile for $\gamma = \pi/4$ and several values of time; **d** magnetization profile at the longest accessible time, $t = 40$, and several values of $\gamma$

energy transport. The magnetization profile is displayed in Fig. 2c for $\gamma = \pi/4$ and several values of time, and in Fig. 2d for several values of $\gamma$ and $t = 40$. Figure 2d undoubtedly shows that, as soon as $\gamma \neq \pi/2$, and thus the right half is described by a Hamiltonian which is different from that of the left half, the magnetization of the left chain deviates from the behaviour of the homogeneous system ($\gamma = \pi/2$) in the region $-0.5 \lesssim n/t < 0$.

Let us start our analysis with a discussion of the regions $0.5 \lesssim |n|/t < 1$, that are identified with the intermediate ballistic regions 2L and 2R. In Fig. 2a, c it is shown that here the energy and magnetization profiles reach a stationary value. In Fig. 2b, d, we compare the long-time behaviour for different values of $\gamma$ in the right half. We observe that in region 2L the energy and magnetization densities are unchanged and coincide with that of the free homogeneous case. This fact is rather surprising, because the energy flow in this region originated by particles emitted from the state in the right half, whose nature does depend on $\gamma$.

We can address this phenomenon by means of a semiclassical and intuitive picture: since this $\gamma$-independent region emerges for sufficiently large rays, it has to be ascribed to the properties of the fastest quasiparticles, that because of their velocity, are essentially transmitted by the junction. However, the fact that the energy profile does not depend on $\gamma$, means that the junction turns them into the fastest carriers of the left half with the same speed. If this interpretation is correct, all the properties of the region $n/t \lesssim -0.5$ (which corresponds to the regions 2L and 1L) should be well-described by the dynamics of the homogeneous case $\Delta = 0$, for which the system is free and exactly solvable.

In order to verify the latter statement we consider a new observable, and in Fig. 3a we present our results for the energy-current profile at some specific values of time, and $\gamma = \pi/4$. A more complete colour-plot of the full space-time of the energy current is provided in Fig. 1, bottom panel. We also superimpose the energy-current profile of the free homogeneous system with

the same initial inverse temperatures $\beta_L \to -\infty$ and $\beta_R \to \infty$ (dashed lines). The predictive power of this simple interpretation of the region 2L is remarkable. Moreover, it is not restricted to the left non-interacting half: considering the homogeneous situation with $\Delta = \cos(\pi/4)$ on the whole chain gives a good description of the current dynamics for $n/t \gtrsim 0.5$ as well (also called regions 2R and 1R). The same statement is true for the case of spin transport, as well. In Fig. 3b, the two 2L and 2R regions are once again well described by the corresponding homogeneous problems initialized in the domain-wall state $|\Psi_2\rangle$ (dashed lines).

We now move to the central regions $0 < |n|/t \lesssim 0.5$, that are identified as the slow relaxation regions 3L and 3R, respectively. In Fig. 3a, b, we observe that around the junction the relaxation dynamics is slow: the current intensity maintains a monotonically decreasing trend even at the longest accessible times of our numerics. The rate of such decay is much slower than the microscopic energy scale $J$ in Eq. (1), and constitutes a second relaxation time scale of the problem (the first one being that necessary to reach steady properties in regions 2L and 2R). Its appearance is due to the existence of the junction and to the fact that translational invariance is locally broken.

It is curious to note that in homogeneous problems where the ballistic transport behaviour is well-described by a hydrodynamic theory, steady properties have been observed to occur *first* around the junction[33,35,36,38]. Here we observe the opposite behaviour: steady properties are first attained close to the edges of the light cone, whereas a slower thermalization process originates at the junction.

We finally analyze how the slow-relaxation regions expand in time. We look at the energy transport setup, but our findings remain valid when looking at the magnetization imbalance as well. In order to quantify their extension, we track the time-evolution of the maximum of the energy current profile in Fig. 3a, which is plotted in Fig. 3c. The emergence of an internal light cone after a transient time is evident. In Fig. 3d, we show the dependence on $\Delta$ of the typical velocity $v_R'$ in the interacting right half; the data are compatible with a linear behaviour passing through the origin. In other words, when $\Delta \to 0$, there is no region of slow dynamics, as expected for the uniform setup. This constitutes an important consistency check of our analysis.

**Thermalization dynamics and stationary transport.** Let us now investigate whether the time-evolved state can be successfully described as a thermal state at temperature $\beta^{-1}$, $\rho \propto e^{-\beta(\hat{H} - h\hat{S}^z)}$. Here an effective magnetic field $h$, acting as Lagrange multiplier, has been introduced in order to take into account that the magnetization along the $z$ axis commutes with the Hamiltonian. We focus for simplicity only on the energy-transport scenario, leaving for brevity the data for spin transport to the Supplementary Note 1. In this case, thanks to the spin-flip invariance $\hat{S}_n^z \to -\hat{S}_n^z$ of the initial state, the Gibbs state is characterized by a zero magnetic field, and only $\beta$ has to be identified.

In practice, we associate an effective inverse temperature $\beta_n^{\mathrm{eff}}(t)$ to each set of three neighbouring sites by considering the local energy density $\varepsilon_n(t)$ for the sites $n$, $n + 1$, $n + 2$ at time $t$, and inverting the thermodynamic relation $\beta(\varepsilon)$ that associates a temperature to the corresponding energy density $\varepsilon$ (see "Methods"). We stress that, by doing so, the inverse temperature $\beta_n^{\mathrm{eff}}(t)$ is both space- and time-dependent. The results are plotted in Fig. 4a, b. We observe that the inverse-temperature profiles recall, in their qualitative features, those of the energy profiles plotted in Fig. 2a, b.

To quantify how well the inverse temperature $\beta_n^{\mathrm{eff}}(t)$ captures the whole set of local properties of the state, which is what

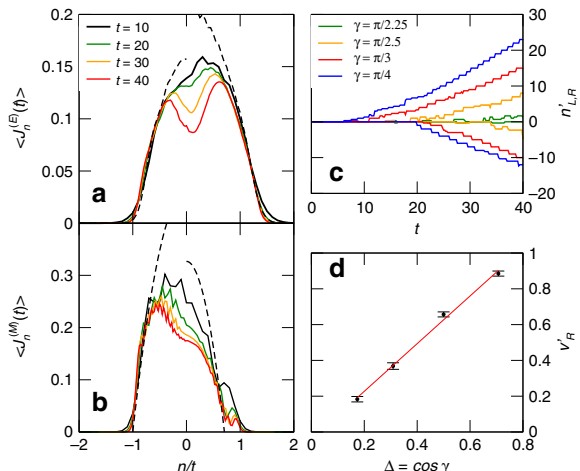

**Fig. 3** Profiles of the energy and magnetization currents during the relaxation process. Energy-transport scenario: **a** energy current profile for $\gamma = \pi/4$ and several values of time. Spin-transport scenario: **b** spin current profile for $\gamma = \pi/4$ and several value of time. In both (**a**) and (**b**), the dashed lines for negative/positive values of $n/t$ are the results of the simulation of an homogeneous systems with the parameters of the left/right halves for $t = 40$, respectively. Internal light cone: **c** space-time plot of the left/right maxima $n'_{L/R}$ of the energy current profile for several values of $\gamma$ (when only one maximum is detected we set $n'_L = n'_R = 0$); **d** extrapolated velocity $v'_R$ of the internal light cone computed by fitting the data of (**c**) for $30 \leq t \leq 40$. The red line is obtained by fitting the data with $v'_R = \mathcal{A}\Delta + \mathcal{B}$ which gives $\mathcal{A} = 1.33 \pm 0.06$ and $\mathcal{B} = -0.03 \pm 0.03$. Error bars represent the standard deviation of the fitting parameters

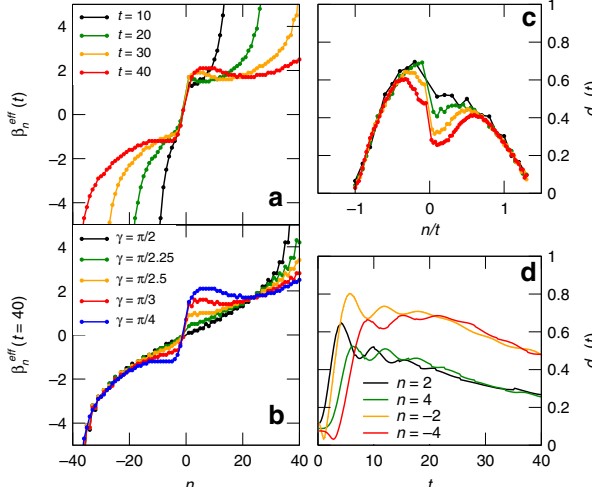

**Fig. 4** Central equilibrium regions in the energy-transport scenario. Effective inverse temperatures: **a** profile of $\beta_n^{\text{eff}}$ for $\gamma = \pi/4$ and several values of time; **b** profile of $\beta_n^{\text{eff}}$ at the longest accessible time, $t = 40$, and several values of $\gamma$. Equilibration tendency: **c** distance between the three-site reduced density matrices $\rho_{n,n+2}(t)$ and $\rho_{n,n+2}[\beta_n^{\text{eff}}(t)]$ for several times and $\gamma = \pi/4$; **d** full time-evolution of the distance for four representative sites taken in both halves for $\gamma = \pi/4$

happens when thermalization occurs, we compute the operator distance between the three-site reduced density matrix of $|\Psi_1(t)\rangle$, dubbed $\rho_{n,n+2}(t)$, and the three-site reduced density matrix of the thermal state $\sim e^{-\beta_n^{\text{eff}}(t)\hat{H}}$, dubbed $\rho_{n,n+2}[\beta_n^{\text{eff}}(t)]$[44]. As an indicator for the distance, we use the trace norm (or Schatten norm for $p = 1$): $d(\rho_1, \rho_2) = \parallel \rho_1 - \rho_2 \parallel_1$, that coincides with the sum of the singular values of the operator difference $\rho_1 - \rho_2$. In particular, we analyze the distance

$$d_n(t) \equiv d\Big( \rho_{n,n+2}(t),\ \rho_{n,n+2}\big[\beta_n^{\text{eff}}(t)\big] \Big). \qquad (2)$$

In Fig. 4c, we plot $d_n(t)$ for $\gamma = \pi/4$ and several values of time. We first focus on the region around the junction. Although a stationary state has not been reached, the pronounced drop of the value of the distance $d_n(t)$ in time hints at a clear tendency towards thermalization. In Fig. 4d, we analyze the time dependence of such distance for four fixed values of $n$, which are taken close to the junction. The plot shows that the relaxation behaviour depends on the considered side of the junction.

The regions $0.5 \lesssim |n|/t \lesssim 1$ do not display any tendency towards thermal equilibrium at all, although here the state relaxes quickly. The original equilibrium regions 1L and 1R, instead, have the minimal value of such distance, and as expected it does not depend on time.

When inspecting in detail the local effective temperatures in Fig. 4a, we observe that two approximate plateaus appear to the left and to the right of the junction, corresponding to regions 3L and 3R. The two plateaus have different values and the interpolation between such them takes place on a length scale of few sites and does not depend on time. The appearance of an almost thermal behaviour around the junction is a distinctive feature of this setup which emerges in the region where a description in terms of ballistic quasiparticles is not possible.

The phenomenology that we just described agrees with the simple picture of a thermal Kapitza boundary resistance: two thermal regions develop around the junction at two different values, and the interpolation takes place on a scale that does not extend with time. Let us stress that a similar situation takes place in the case of magnetization transport, and in that case we can

speak of a magnetic Kapitza boundary resistance. This phenomenology has been widely observed in classical 1D systems and several mesoscopic transport experiments[5,6], while here it is first presented in a fully-quantum description.

The fact that two regions with different temperatures are put closed by poses the question of whether a stable energy current develops between them. We explore this problem in Fig. 5a, b, where we plot the time-dependence of $\langle \hat{J}_{n=0}^{(E)} \rangle(t)$ and $\langle \hat{J}_{n=0}^{(M)} \rangle(t)$ (the latter plot refers to the case of magnetic transport); different values of $\Delta$ are considered. In both cases, the current reaches a steady value for $\Delta = 0$ ($\gamma = \pi/2$) but, as soon as $\Delta \neq 0$, it shows a slow decreasing trend; the effect becomes more pronounced with increasing $\Delta$.

This plot poses a problem because we just pointed out that two regions with different temperatures build up close by, and we would thus expect a saturation of energy or magnetic current with time, even in the cases where $\Delta \neq 0$. It is thus an interesting question to assess the long-time limit of our model. Yet, our numerics witnesses an uninterrupted decaying trend. With the numerical methods at our disposal, we cannot make a conclusive statement about the longer-time behaviour of such current. However, in the next section we will argue that at the junction an homogeneous thermal state cannot be created: therefore, a finite current is generically expected to persist in the stationary state.

We conclude our numerical analysis on the thermalization dynamics, by arguing that also bipartite entanglement[45,46] witnesses a qualitative difference between the homogeneous and non-homogeneous cases. We take advantage of the fact that our system is pure, so that the von Neumann entropy of the reduced density matrix of the left part $\rho_{\text{left}}(t) = \text{Tr}_{\text{right}}[|\Psi_i(t)\rangle\langle\Psi_i(t)|]$ is a good measure of the entanglement between the left and the right half at each time. Its mathematical expression reads $S(t) = -\text{Tr}\big[\rho_{\text{left}}\ln(\rho_{\text{left}})\big]$.

In Fig. 5c, d, we plot $S(t)$ for several values of $\gamma$. Both in the case of energy and spin transport, the growth of $S(t)$ is logarithmic for the homogeneous case $\gamma = \pi/2$: this is a consequence of integrability, which allows evolving the initial state with a weak generation of entanglement on each ray[47–52]. Indeed, as soon as the system becomes inhomogeneous, the behaviour of the entropy qualitatively changes and grows super-logarithmically, testifying the particular process that is taking place at the junction. The data are particularly clear for the case of spin transport, where the initial state is an uncorrelated product state.

**Absence of complete thermalization at the junction**. In order to explain the slow and peculiar dynamics around the junction, a natural possibility is that the system is undergoing full thermalization; such a scenario would be motivated by the fact that integrability is always broken for $\gamma \neq \pi/2$[27]. However, it is undeniable that, at the time scales accessible with our numerics, relaxation did not yet fully take place. We now discuss how the numerically observed scenario may evolve at later times. In the following discussion, the thermodynamic limit is assumed, or analogously that $L \gg \max[v_L t, v_R t]$, so that the boundary plays no role.

The relaxation dynamics of closed many-body quantum systems has been thoroughly explored in the last decade[8–12]. Several studies in homogeneous settings have shown that equilibration generically occurs, and that the stationary state is a generalized Gibbs ensemble (GGE). When the model is integrable, it must take into account the full set of local conserved quantities of the model. Since our model is not integrable, the corresponding ensemble is solely characterized by temperature and magnetic field. However, deep in the bulk of each half and far

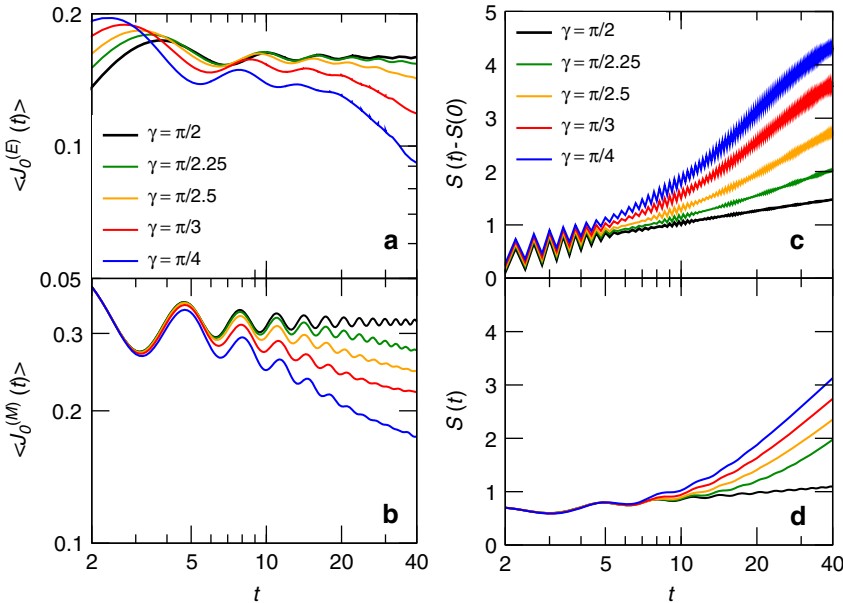

**Fig. 5** Behaviour at the junction. Decay of the current at the junction: **a** energy-transport scenario; **b** spin-transport scenario. Time-dependence of the entanglement entropy of a system bipartition: **c** energy-transport scenario; **d** spin-transport scenario

from the junction, the former approach could be applied also to our model. We now show that this possibility forbids a complete thermalization of the system at the junction and spreading from the junction. In particular, we demonstrate that it is not possible to fully describe the junction with a single well-defined temperature, and thus to assume that no current flows there.

We introduce the concept of rapidity $\zeta = x/t$ and consider the limit of infinite times at fixed rapidity $\zeta$. We study the expectation value of a generic local observable $\hat{\mathcal{O}}_n$ at fixed rapidity:

$$\lim_{t\to\infty}\langle\Psi(t)|\hat{\mathcal{O}}_{n\sim\zeta t}|\Psi(t)\rangle = \mathrm{Tr}[\rho_\zeta\hat{\mathcal{O}}]. \quad (3)$$

Here $\rho_\zeta$ characterizes the LQSS, the density matrix on each ray of the space-time with rapidity $\zeta$.

Since the state is integrable within each half, we assume that for any $\zeta<0$ ($\zeta>0$), $\rho_\zeta$ takes the form of a GGE expressed in terms of the left (right) conserved quantities. This observation is not sufficient to identify the specific form of GGE; however, using the integrability of each half, we obtain that the expectation value in Eq. (3) should be a smooth function of $\zeta$. This is sufficient to uniquely identify $\rho_\zeta$ once the appropriate boundary conditions for $\zeta \to 0^\pm$ and for $\zeta \to \pm\infty$ are imposed (see Methods for more details).

Clearly, if $\zeta<v_L$ or $\zeta>v_R$, we are inspecting the state outside the causal region; since here no dynamics can occur, $\rho_\zeta$ must coincide with the initial state. Instead, the properties at $\zeta=0$ refer to any fixed but arbitrary distance $n$ from the junction. Let us now assume that the junction lets the whole system thermalize:

$$\lim_{t\to\infty}\langle\Psi(t)|\hat{\mathcal{O}}_n|\Psi(t)\rangle = \mathrm{Tr}[\hat{\mathcal{O}}_n\rho_{\mathrm{TH}}], \quad (4)$$

$$\rho_{\mathrm{TH}} = \frac{e^{-\bar{\beta}(\hat{H}-\bar{h}\hat{S}^z)}}{Z}. \quad (5)$$

Here $Z$ is the partition function while $\bar{\beta}$ and $\bar{h}$ are the uniquely defined effective inverse temperature and magnetic field, respectively. This is equivalent to ask that for $\zeta=0$ a thermal state must emerge at large times:

$$\rho_{\zeta=0} = \rho_{\mathrm{TH}}. \quad (6)$$

In practice, the density matrix $\rho_\zeta$ must interpolate smoothly between the initial states outside the light cone and the thermal state at the junction. In particular:

$$\lim_{\zeta\to0^-}\rho_\zeta = Z^{-1}e^{-\bar{\beta}\sum_{n<0}(\hat{h}_n-\bar{h}\hat{S}^z_n)}, \quad (7)$$

$$\lim_{\zeta\to0^+}\rho_\zeta = Z^{-1}e^{-\bar{\beta}\sum_{n>0}(\hat{h}_n-\bar{h}\hat{S}^z_n)}. \quad (8)$$

We stress that the two limits in Eqs. (7) and (8) are two different states because the local Hamiltonian $\hat{h}_n$ defined in Eq. (1) depends on the sign of $n$. Yet, there is one unique temperature and magnetic field that eventually describe the whole system.

We now show that the situation described by Eq. (6) is impossible because one such solution is incompatible with the free propagation of quasiparticles within each half. Let us consider for simplicity the left half, where $\Delta = 0$, and consider through Jordan-Wigner transformation its fermionic quasiparticles, that are uniquely characterized by their momentum $k \in [-\pi,\pi]$. For a negative rapidity $\zeta$, since the system is non-interacting, in order to specify $\rho_\zeta$ as a GGE, we have to specify the occupation number of the quasiparticles, $n_\zeta(k)$. At large times on this ray, the movers with velocity smaller than $\zeta$ will be coming from the junction described by $\rho_{\mathrm{TH}}$; instead, the movers with velocity larger then $\zeta$ will have propagated almost freely from the initial state on the left:

$$n_\zeta(k) = \begin{cases} n_{\mathrm{TH}}(k) & v(k)<\zeta, \\ n^{(L)}(k) & v(k)>\zeta. \end{cases} \quad (9)$$

Here $n_{\mathrm{TH}}(k)$ is the quasi-particle occupation number associated to $\rho_{\mathrm{TH}}$, whereas $n^{(L)}(k)$ is the one associated to the original state on the left. One such LQSS is not thermal because in the thermal case all movers should be characterized by the same temperatures. This remains true taking the limit $\zeta \to 0^-$. Thus, the corresponding LQSS at $\zeta \to 0^-$ will not be a thermal state and Eq. (6) is violated (see "Methods" for more details and for the proof of Eq. (9)).

Therefore, we expect that at large times, even though integrability is broken, the system will equilibrate to a non-thermal steady state. As a first consequence, it is not possible to exclude a priori that an energy current will steadily flow at long

time, and that the temperature discontinuity observed in the numerics will always be present around the junction. This is the phenomenology that we previously identified with a Kapitza boundary resistance.

If we assume that, as suggested by our numerics, thermal regions develop close to the junction, our results point at the existence of a finite region around the junction that connects the $\zeta = 0^-$ and $\zeta = 0^+$ GGEs without widening in time. This latter scenario is consistent with those of refs. [29,42] in a related setting.

**A solvable model for the junction.** In the previous section, we argued that the microscopic details of the junction determine how the GGE on the left is connected to the one on the right, and that this connection takes place in a finite region that does not scale with time. This is what we observe in our numerics at finite times. In this section we present a simplified junction model which analytically supports that such scenario can survive at infinite times. As we will discuss, we do not expect the model to reproduce quantitatively the numerical data, but to provide a simple framework to understand the temperature discontinuity at the boundary of the junction.

Instead of directly connecting the left and right Hamiltonians, we insert a finite region of length $\ell$ centred around $n = 0$, whose Hamiltonian $\hat{H}$chaos is fully chaotic (see Fig. 6). Such Hamiltonian is not integrable, is disordered, and features diffusive transport, as in ref. [43]. The idea is that locally the system relaxes in finite (and not extensive) time scales $\tau_{\text{rel}}$, and that $\tau_{\text{rel}} \ll \ell^2/D$, with $D$ the diffusion constant. We still assume that the total magnetization is conserved, i.e., $[\hat{H}\text{chaos}, \hat{S}^z] = 0$, though for

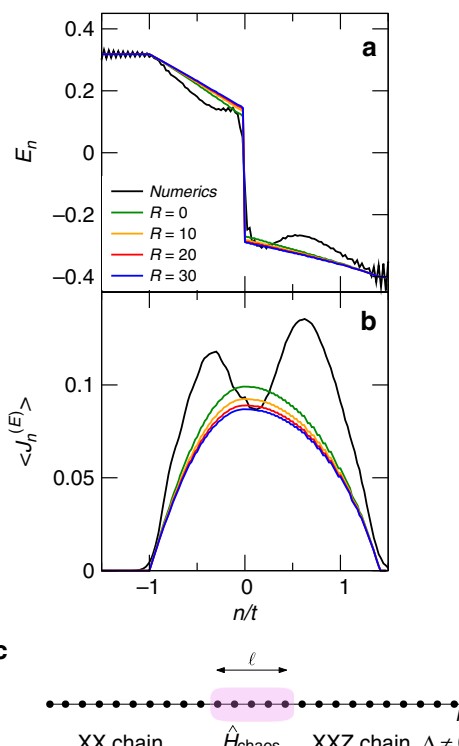

**c**

XX chain    $\hat{H}_{\text{chaos}}$    XXZ chain, $\Delta \neq 0$

**Fig. 6** Extended chaotic junction. **a, b** Analytic results from GHD for $\gamma = \pi/4$ in the energy-transport scenario for different values of the boundary resistance $R$, as indicated in the legend; **a** energy profile; **b** energy current. The value $R = 0$ corresponds to a limiting situation where $|T_- - T_+| \to 0$ but $(T_- - T_+)/R$ is finite. The black line shows the numerical data for $t = 40$. **c** Cartoon of the setting where the two integrable Hamiltonians are joined via a chaotic Hamiltonian $\hat{H}_{\text{chaos}}$ on $\ell$ sites

simplicity we will only focus on energy transport. The choice of a finite junction is motivated qualitatively by our numerical observations, although the model that we are here discussing requires $\ell$ to be much larger than the few sites considered numerically.

The argument of the previous section still applies, so that no overall thermalization shall be expected. Nevertheless, if the value of $\ell$ is large, we can once again employ a hydrodynamic description. Taking a coarse-grained coordinate $x \in [-\ell/2, \ell/2]$, we assume local equilibration to a thermal ensemble characterized by a space-time dependent temperature $T(x, t)$. Then, the two fundamental equations describing temperature and energy-current dynamics in this region follow from the energy conservation and Fourier's law, and read:

$$\partial_t[s(T)T(x, t)] = -\partial_x[J_E(x, t)], \tag{10}$$

$$J_E(x, t) = -\kappa(T)\partial_x[T(x, t)], \tag{11}$$

where $s(T)T(x, t)$ is the energy density, $s(T)$ being the volumetric heat capacity, whereas $\kappa(T)$ is the thermal conductivity. In this hydrodynamic picture, the microscopic details of $\hat{H}$chaos are encoded in $s(T)$ and $\kappa(T)$; for weak temperature variations, we can assume that they are constant, namely $\kappa(T) = \kappa_0$ and $s(T) = s_0$. In the stationary limit, the temperature profile depends linearly on $x$ and the energy current is uniform. The stationary value of $J_E$ depends on the temperatures at the edge of the junction $T_\pm$:

$$T_\pm = \lim_{t \to \infty} T(\pm \ell/2, t), \tag{12}$$

$$\lim_{t \to \infty} J_E(x, t) = \kappa_0(T_- - T_+)/\ell. \tag{13}$$

This last equation provides the definition of a thermal Kapitza boundary resistance:

$$R = \frac{\ell}{\kappa_0}, \tag{14}$$

We stress that, although we use the notion of temperature to describe the region around the junction, this is not invalidating the results of the previous section. Here, a current is flowing and thus thermalization did not completely take place; in this context, the notion of a temperature is thus an approximation holding in a hydrodynamic regime, which becomes more and more accurate as $\ell \to \infty$.

We use GHD to describe energy transport in the two halves for any non-vanishing rapidity $\zeta = x/t$ and bridge the original properties outside the light cone with the obtained solution around the junction. In practice, one looks for a solution in the left and right half that (i) has the proper behaviour outside the light-cone, and (ii) has the same energy current [see Eq. (13)] in the limit $\zeta \to 0^\pm$. Therefore, once the parameter $R$ is fixed for the junction model, the temperatures $T_\pm$ and the stationary state for each rapidity are determined.

In Fig. 6, we plot the energy and current profile for different values of the resistance $R$ for the initial state $|\Psi_1\rangle$ obtained with the chaotic junction model. The energy profile has a discontinuity; this clearly shows that the Kapitza boundary scenario can be reproduced by an analytical model addressing time-scales longer than those of our numerics.

The model has not been introduced to explain the numerical data in Figs. 2 and 3; nonetheless we investigate whether it can also approximately describe them. The aforementioned discontinuity in the energy profile is reminiscent of the behaviour observed in Fig. 4; for a better readability, we superimpose the numerical data for $\gamma = \pi/4$ at the longest accessible time. A qualitative agreement is observed.

The energy current is continuous by construction, but in this case the agreement with the numerics is poor. In particular, in the analytical solution we do not observe two different regions inside the causal cone; but rather a unique one. We understand the distinction between regions 2 and 3 in our numerics as due to the different dynamics of fast and slow excitations. However, in the considered limit of a large chaotic junction, all quasiparticles undergo a large number of scattering events, irrespectively of their velocity. Thus, we ascribe the two different behaviours to the specific properties of the two junction models.

We conclude by mentioning that the energy current measured in our numerics still shows, at the longest accessible times, a significant decreasing trend. This is in contrast with the long-time predictions of the chaotic junction model and suggests a long relaxation time, as well as a large resistance. Understanding the microscopic origin and generality of this phenomenology are interesting questions that cannot be discussed with present numerical tools, and are thus left as an open issue for future investigations.

## Discussion

Ballistic transport of particles and energy in 1D quantum systems is a fundamental concept in non-equilibrium physics. This phenomenology has been observed experimentally in a wide spectrum of physical platforms ranging from nanowires and carbon nanotubes to the edge states of Hall bars and topological insulators. In this work we addressed the problem of studying the emergence of Kapitza boundary resistances when two 1D quantum ballistic conductors are joined together.

Recent works have pointed out that transport phenomena are tightly bound to a number of fundamental issues in statistical mechanics. The study of equilibration in 1D closed quantum systems has recently assessed that homogeneous integrable models have a peculiar long-time dynamics because in these systems quasi-particles interact without suffering from back-scattering. This allows for the presence of stationary ballistic currents and is accounted for by the GHD description of ballistic transport.

The situation becomes less trivial when the quasiparticles of the two conductors have different nature[27,53]. From a theoretical viewpoint the system becomes non-integrable and the known framework does not apply any longer. Yet, ballistic transport (a consequence of integrability) is supported in each separate half of the chain. In general, the community is currently considering the presence of forms of local integrability breaking terms, such as localized defects[27,29,42,54–56]; our work fits within this research effort.

Although our system is not integrable, our conclusions are in sharp contrast with what one would expect for generic chaotic and diffusive systems, even when an inhomogeneous background is considered[57]. For instance, the light cone characterized by velocities $v_{L,R}$ is due to the integrability of each half and is not expected to appear in fully non-integrable models, where rather a diffusive scenario is the anticipated situation. Moreover, our numerics witnesses in real time the appearance of a slow-relaxing region that expands from the center. This region is the direct consequence of the integrability-breaking properties of the junction. At asymptotically long times, we argue that the system eventually reaches a state which supports stationary currents even though integrability is broken. The two halves equilibrate to distinct GGEs, which are connected in a non-trivial way by the junction.

We conclude by stating that systems where integrability is broken only in a finite region of space are therefore special, even though they look non-integrable under conventional indicators

such as the level spacing statistics[27,29]. This demands a novel characterization of integrability (or the lack thereof) appropriate for inhomogeneous settings.

This work may be the object of experimental investigations in several platforms where coherent Hamiltonian dynamics can be easily realized, and the partitioning protocol employed in this work might reveal crucial for studying transport phenomena in systems where leads are not easily implemented. A natural possibility are ultra-cold atoms, where the realisation of 1D lattice systems is possible and recent in-situ microscopy techniques allow for a site-resolved study of coherent time evolution[58]. The recent developments of assembled quantum simulators motivate similar investigations in arrays of Rydberg atoms[59]. Superconducting circuits have recently attained significant coherence times, and can realize one-dimensional systems with tunable (and in particular inhomogeneous) non-linearities[60]. Here too our findings could be investigated.

## Methods

**Details on numerical simulations.** The numerical simulations of the time evolution of the quantum spin chain are performed by means of a matrix product state (MPS) ansatz for the wavefunction[13]:

$$|\Psi\rangle = \sum_{\sigma_1,\sigma_2\dots\sigma_N} c_{\sigma_1,\sigma_2\dots\sigma_N} |\sigma_1,\sigma_2\dots\sigma_N\rangle; \tag{15}$$

$$c_{\sigma_1,\sigma_2\dots\sigma_N} = A^{[\sigma_1]}_{\alpha_1} A^{[\sigma_2]}_{\alpha_1,\alpha_2} \cdots A^{[\sigma_{N-1}]}_{\alpha_{N-2},\alpha_{N-1}} A^{[\sigma_N]}_{\alpha_{N-1}}. \tag{16}$$

Summation over repeated $\alpha$ indexes is assumed. The matrices $A^{[\sigma_j]}$ have two indexes which label the bond link and can take at most $\chi$ values (the first and the last one are exceptional and have only one index, in order to enforce open boundary conditions). This latter parameter sets the precision of the ansatz and of the associated numerical simulation. Roughly speaking, the larger the bond link, the more correlations within the state can be faithfully described; a small bond link, instead, describes an almost uncorrelated state ($\chi = 1$ is a product state).

The first step requires the preparation of one of the initial states $|\Psi_1\rangle$ or $|\Psi_2\rangle$, detailed in the text. The former is prepared by a standard iterative ground-state search[13] of Hamiltonian (1) with $\lambda_n = -1$ for $n \leq 0$ and $\lambda_n = 1$ for $n > 0$. The latter is a MPS with bond-link dimension $\chi = 1$ which is constructed explicitly.

The system is subsequently evolved in time with the Hamiltonian $\hat{H} = \sum_n \hat{h}_n$, where $\hat{h}_n$ is given in Eq. (1). We employed the time-evolving block-decimation (TEBD) algorithm with a fourth-order Trotter decomposition and time step $dt = 0.02J^{-1}$. We have checked that numerical inaccuracies due to this choice are negligible on the scales of all the figures shown here.

The bond-link dimension $\chi$ of the MPS representation increases exponentially in time because of the spreading of correlations. Furthermore, the fact that the system is inhomogeneous increases the complexity of the state, as already observed by a direct comparison of the results published in Refs. [27,61]. Since a higher bond-link dimension implies a larger computational cost to evolve the MPS, an upper bound $\chi_{\max}$ of the representation needs to be set. Remarkably, in all the simulations performed here, convergence for observables under consideration for $t \leq 40$ has been reached for $\chi_{\max} \leq 500$. In order to faithfully explore the large times considered in the paper we simulated a chain made of $L = 140$ sites with open boundary conditions. This guarantees that, for the parameters considered in this work, the dynamics is not affected by the boundaries.

**Effective local temperature and chemical potential.** In order to ascribe an effective local temperature $\beta^{\text{eff}}_n$ to the time-evolved state, one needs to compare the local energy of the system at time $t$ with that of a system at thermal equilibrium. This quantity is both space (since our setup is not homogeneous) and time dependent. Let us stress that the state $|\Psi_1\rangle$ does not involve any spin transport since $\langle\Psi_1|\hat{S}^z_n|\Psi_1\rangle = 0$, $\forall n$ and the Hamiltonian specified by Eq. (1) conserves the total magnetization. As a consequence, we can restrict our analysis to states with zero effective local chemical potential.

At each time $t$, we may define the local energy at the site $n$ by considering a subsystem of three contiguous sites $\{n, n+1, n+2\}$. The average energy is thus given by $\overline{E}_n(t) = (\langle\hat{h}_n\rangle_t + \langle\hat{h}_{n+1}\rangle_t)/2$, where $\langle\cdot\rangle_t$ denotes the expectation value over the time-dependent state. Next, we consider a homogeneous system with the parameters of the left (if $n \leq 0$) or right half (if $n > 0$) at thermal equilibrium, and compute the average energy density in the bulk $E(\beta)$; note that $\beta$ can be both positive and negative. Inverting this relation, we can associate to the system a local effective inverse temperature $\beta^{\text{eff}}_n$, for each time of the time evolution.

The reconstruction of the three-site reduced density matrix $\rho_{n,n+2}(t)$ requires the calculation of 64 observables and has been performed only for $\gamma = \pi/4$ and $\gamma = \pi/2$. The latter is the homogeneous and non-interacting case and is used as a

benchmark to highlight the peculiarities of our inhomogeneous setups (see Supplementary Note 2 for details). For all other values of $\gamma$ plotted in Fig. 4b, the average energy is simply computed as $E_n(t) = \langle \hat{h}_n \rangle_t$.

The non-equilibrium dynamics generated from the *domain-wall* state $|\Psi_2\rangle$ is characterized by a non vanishing local energy $E_n(t)$ and magnetization $S_n^z(t) = \langle \hat{S}_n^z \rangle_t$. Therefore, in addition to the local inverse temperature $\beta_n^{\mathrm{eff}}(t)$, we may identify an effective local chemical potential $\mu_n^{\mathrm{eff}}(t)$ as well. For the gapless XXZ chain at finite temperature and chemical potential, these quantities can easily worked out by standard thermodynamic Bethe ansatz (TBA)[33,62].

We thus proceeded as follow: (i) for fixed $\beta$ and $\mu$, we associated to each point in space a TBA description in terms of root densities $\{\vartheta_n(\lambda)\}$ depending on the local value of the anisotropy $\Delta$. (ii) we computed, in that local TBA, both the energy density $e_n(\beta, \mu)$ and the magnetization density $s_n(\beta, \mu)$ for $\beta \in [-5, 5]$, $\mu \in [-5, 5]$ and discretization step $d\beta = d\mu = 0.1$; (iii) after interpolating those functions, we were able to extract the time-dependent local effective temperature and chemical potential by numerically solving the two equations $e_n(\beta, \mu) = E_n(t)$ and $s_n(\beta, \mu) = S_n^z(t)$.

**Details on the impossibility of a complete thermalization.** We now provide some technical details on the proof of the absence of complete thermalization. We first focus on the left half of the system. As the model is integrable in the bulk, in the thermodynamic limit it admits an infinite and complete set of conserved densities $\{\hat{q}_\mu^j\}_{\mu=1}^\infty$ [63], whose support is localized around the site $j$ (possibly with exponential tails[64]). Each density satisfies the operatorial continuity equation

$$\mathrm{i}[\hat{H}_L, \hat{q}_\mu^j] = \hat{J}_j^\mu - \hat{J}_{j+1}^\mu, \tag{17}$$

provided that $j \leq j^*$, where $j^*$ negative and sufficiently distant from the junction. Now consider the total amount of a given conserved quantity in the region of space $[-\zeta t, j^*]$

$$Q_\mu(t; \zeta) = \sum_{j=-\zeta t}^{j^*} \mathrm{Tr}[\hat{q}_\mu^j(t)\rho_0]. \tag{18}$$

Using Eq. (17), the time derivative $\partial_t Q_\mu(t; \zeta)$ involves the expectation value of the current $\mathrm{Tr}[\hat{J}_j^\mu \rho_0(t)]$ at the endpoints $j = j^*$ and $j = -\zeta t$, which can be computed using Eqs. (3) and (4). A mismatch between the stationary currents computed with $\rho_{\mathrm{TH}}$ and $\rho_\zeta$ would imply a finite rate of growth of the charge $Q_\mu(t; \zeta)$ in the space region $[-\zeta t, j^*]$, which however is always bounded by $||\hat{q}_\mu||_\infty \times \zeta$ with $|| \cdot ||_\infty$ the operator norm and $\zeta$ arbitrarily small. Repeating the argument for $x > 0$, it follows that

$$\mathrm{Tr}[\rho_{\mathrm{TH}} \hat{J}^\mu] = \lim_{\zeta \to 0^\pm} \mathrm{Tr}[\rho_\zeta \hat{J}^\mu]. \tag{19}$$

As Eq. (19) must hold for any $\mu$, it imposes an infinite set of constraints on the GGE density matrices $\lim_{\zeta \to 0^\pm} \rho_\zeta$. For the XX model, this is enough to deduce Eq. (7), because completeness of densities $\{\hat{q}_\mu^j\}_\mu$ implies the one of currents[65]. On the contrary, in the presence of interactions, such an inference is subtler. For integrable models supporting relativistic invariance, densities and currents are related by Lorentz transformation, Eq. (19) is sufficient to deduce Eqs. (7) and (8). Here, following the arguments which led to GHD in[32], we assume that Eqs. (7) and (8) must hold whenever Eq. (19) holds.

However, the ballistic propagation of quasiparticles within each half is generically incompatible with Eqs. (7) and (8). For the sake of simplicity we analyse the XX case. After a Jordan–Wigner transformation, the model is diagonalized by fermionic operators in Fourier space $\hat{c}_k, \hat{c}_k^\dagger$, satisfying $\{\hat{c}_k^\dagger, \hat{c}_{k'}\} = \delta_{k,k'}$. Then, a GGE state is completely characterized by the occupation number $n(k) = \mathrm{Tr}[\hat{c}_k^\dagger c_k \rho_0(t)]$ at momentum $k$ and we indicate with $n^{(L)}(k)$ the state corresponding to $\rho_L$. Then, according to GHD, the occupation number is promoted to a space-time dependent function satisfying[28,66,67]

$$\partial_t n(k; x, t) = v(k)\partial_x n(k; x, t). \tag{20}$$

Here, the velocity takes the simple form $v(k) = d\epsilon(k)/dk$ with the dispersion relation $\epsilon(k) = -J\cos(k)$. Equation (20) needs to be solved in the domain $x \in (-\infty, 0)$, with the boundary conditions: $n(k; x < 0, t = 0) = n^{(L)}(k)$ and $n(k; x = 0, t) = (1 + e^{\bar{\beta}(\epsilon(k) - \bar{h})})^{-1} \equiv n_{\mathrm{TH}}(k)$, the equilibrium Fermi-Dirac distribution at temperature $\bar{\beta}$ and chemical potential $\bar{h}$. The solution in the limit $x/t = \zeta \to 0^-$ is easily found to be:

$$n(k; x, t) \overset{x/t \to 0^-}{=} \theta(-v(k))n_{\mathrm{TH}}(k) + \theta(v(k))n^{(L)}(k). \tag{21}$$

Equations (7) and (8) are clearly not satisfied, except for the trivial case $n^{(L)}(k) = n_{\mathrm{TH}}(k)$ where no dynamics occurs. In a similar way, one can extend the same argument to the interacting integrable case for $x > 0$[68,69]. This shows that Eq. (4) is inconsistent with the ballistic propagation of quasiparticles.

**Details on the chaotic model of the junction.** In this section, we provide additional details about the model of diffusive junction and how the data in Fig. 6 were obtained. As explained in the previous section, integrability of the two halves implies that they admit a description in terms of GHD. For the left half, Eq. (20) applies. Instead, for the right half, the XXZ model with $\Delta \neq 0$ requires Eq. (20) to be generalized as[33]

$$\partial_t \vartheta_\alpha(\lambda; x, t) = v_\alpha(\lambda; \vartheta)\partial_x[\vartheta_\alpha(\lambda; x, t)]. \tag{22}$$

Comparing with Eq. (20), we changed the notation for the occupation number $n \to \vartheta_\alpha$ and we employed a parametrization in terms of the rapidity $\lambda$ instead of the momentum $k$ in agreement with standard literature. For $\Delta = \cos(\pi/P)$, the index $\alpha = 1, \dots, P$ labels the particle type and is due to the presence of stable bound-states (see ref. [62] for generic values of $\Delta$). A fundamental difference is that the velocity is dressed by the interactions and becomes a state-dependent quantity $v_\alpha(\lambda; \vartheta) = [\epsilon']_\alpha^{\mathrm{dr}}(\lambda)/[k']_\alpha^{\mathrm{dr}}(\lambda)$[34], where $\epsilon_\alpha(\lambda)$ and $k_\alpha(\lambda)$ are respectively the bare single-particle energy and momentum. The dressing $q(\lambda) \to [q]^{\mathrm{dr}}(\lambda)$ is a linear-operation defined for any set of functions $q_\alpha(\lambda)$ of rapidities via

$$[q]_\alpha^{\mathrm{dr}}(\lambda) = q_\alpha(\lambda) + \sum_\beta \int d\mu \, \mathrm{T}_{\alpha,\beta}(\lambda,\mu)\vartheta_\beta(\mu)[q]_\beta^{\mathrm{dr}}(\mu), \tag{23}$$

The kernel $\mathrm{T}_{\alpha,\beta}(\lambda,\mu)$ depends on the interaction and we refer the reader to[33,35] for the explicit formulas as a function of $\Delta$. Every GGE density matrix $\rho_{\mathrm{GGE}}$ is characterized by the set of functions $[\vartheta_\alpha(\lambda)]_{\alpha=1}^P$ [35,62–64]. For instance, the expectation value of a conserved density $\hat{q}^j$ and current $\hat{J}^j$ are written as

$$\mathrm{Tr}[\rho_{\mathrm{GGE}} \, \hat{q}^j] = \sum_\alpha \int \frac{d\lambda}{2\pi} \, [k']_\alpha^{dr}(\lambda)\vartheta_\alpha(\lambda)q_\alpha(\lambda), \tag{24}$$

$$\mathrm{Tr}[\rho_{\mathrm{GGE}} \hat{J}^j] = \sum_\alpha \int \frac{d\lambda}{2\pi} [k']_\alpha^{dr}(\lambda)\vartheta_\alpha(\lambda)v_\alpha(\lambda)q_\alpha(\lambda), \tag{25}$$

where $q_\alpha(\lambda)$ are the single-particle eigenvalues corresponding to the conserved density $\hat{q}^j$.

In the study of thermal transport, Eqs. (20), (22) in the region $|x| > \ell/2$ can be used in combination with Eq. (11) in order to have a full hydrodynamic description. The initial state $|\Psi_1\rangle$ corresponds to the left/right GGE states[62]

$$n^{(L)}(k) = \theta(|k| - \pi/2) , \qquad \vartheta_\alpha^{(R)}(\lambda) = \delta_{\alpha,1} . \tag{26}$$

where $\theta(x)$ is the Heaviside step function. In practice, Eqs. (20), (22) can be solved in the domain $|x| > \ell/2$, while the junction acts as a source of thermal particles at $x = \pm \ell/2$ respectively at the instantaneous temperatures $T(\pm \ell/2, t)$. In the simplified case of constant diffusivity $\alpha_0$ and conductivity $\kappa_0$, we can directly look for a stationary solution at $t \to \infty$. In this case, the stationary temperatures $T_\pm$ in Eq. (12) at the edges of the junction are the only unknowns. One can then solve Eqs. (20) and (22) at fixed rapidities $x/t = \zeta$[33]:

$$n(k; x, t) = \theta(\zeta - v(k))n^-(k) + \theta(v(k) - \zeta)n^{(L)}(k) , \tag{27}$$

$$\vartheta_\alpha(\lambda; x, t) = \theta(v_\alpha(\lambda) - \zeta)\vartheta_\alpha^+(\lambda) + \theta(\zeta - v_\alpha(\lambda))\vartheta_\alpha^{(R)}(\lambda) . \tag{28}$$

where $n^-(k)$ and $\vartheta_\alpha^+(\lambda)$ are the thermal occupation numbers for the left and right models, respectively at temperatures $T^-$ and $T^+$. Then, one can determine the values of $T_\pm$ by imposing that the left/right energy currents computed via Eq. (25) at $\zeta = 0^\pm$ match with Eq. (13) and the only phenomenological parameter is the Kapitza resistance in Eq. (14).

The spin transport from the state $|\Psi_2\rangle$ can be modeled in a similar way. However, in this case the initial magnetization imbalance induces also a non-vanishing energy current. A model of a diffusive junction in this case would require a $2 \times 2$ Onsager matrix of phenomenological coefficients. For simplicity, we do not consider this case here.

## Data availability
Data are available upon reasonable request from the authors.

## Code availability
All numerical codes in this paper are available upon reasonable request to the authors.

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

## Acknowledgements

We thank A. Bastianello, J. De Nardis, M. Fagotti, R. Fazio and A. Nahum for enlightening discussions. We acknowledge support from the BMBF and EU-Quantera via QTFLAG and the Quantum Flagship via PASQuanS, the European Unions Horizon 2020 research and innovation programme under the Marie Sklodowska-Curie Grant Agreement No. 794750 (A.D.L.).

## Author contributions

All the authors conceived the work, agreed on the approach to pursue, analysed and discussed the results; A.B. and M.C. performed the numerical simulations; A.D.L., L.M. and D.R. supervised the work.

## Competing interests

The authors declare no competing interests.
