## [Peer Review File · Nature Communications]

Reviewers' comments:

Reviewer #1 (Remarks to the Author):

The manuscript of Biella et al. present an interesting numerical experiment on heat and magnetization transport in inhomogeneous spin chains, composed of two semi-infinite pieces of integrable chains, specifically a free (XX) chain and an interacting XXZ chain.

The authors argue that, despite manifest integrability breaking, the non-equilibrium transport in the model is not compatible with the assumption of thermalization. Specifically, heuristic and semi-analytical arguments are given that support ballistic quasiparticle transport sufficiently far from the junction but still well inside the Lieb-Robinson light-cone.

I like the results and the reasoning of the paper. I think as well that this is a very interesting and important problem. That being said, I believe that the interpretation of the results is not unambiguous, and that the paper does not (and perhaps can not) make a very clear conclusion.

For example, the chaotic-impurity-region model assumes that the size of impurity-region ℓ does not grow with time. Under this assumption one easily obtains the Kapiza boundary resistance model, Eq. (7), and the resulting transport remains ballistic. But this model would then also predict that the current $J(n=0,t)$ would NOT decay as a function of t , right? How can this be reconciled with Fig. 6, where, say for $\gamma=\pi/4$, we notice decay of the central-bond current after times $t\sim 10-20$.

I believe this issue will be hard to decide based on the available data which are probably state-of-the-art with the TEBD method. But at least a clear discussion of this seeming inconsistency should be in order. And perhaps commenting why authors believe that ℓ will not grow with t (perhaps for some small-enough growth one has no inconsistency with their no-go statement for thermalizing junction)?

Some more specific remarks about the manuscript:

- Page 2, left column: The author state that "Pure states are numerically more tractable than density matrices and allow us to explore longer time dynamics". This is in contrast to the statement of a closely-related paper Nature Comm. 8, 16117 (2017) - which I believe should be cited - where the authors advocated density-matrix TEBD, at least in the linear response regime, as superior to pure-state TEBD in exactly the same context. The resolution of the dilemma is probably in the fact that the authors here are interested in very *specific* pure initial states, namely such as corresponding to ground (or extremal) eigenstates of the half-chains. For more generic pure (even though possibly separable) initial state, the situation would be very different. This is also clear from very moderate scaling of the entanglement entropies shown in Fig. (6). Also a comment on how different would be a regime of linear response, where one would make a slight step in β or magnetization across the junction, would be useful.

- Language is sometimes quite convoluted and consequently presentation is difficult to follow. For example, two sentences in the second paragraph, right column, page 2, starting with "Let us stress that..." were difficult to comprehend. I would suggest to replace the phrase "around the junction" with something more concise.

- In the argument about impossibility of a complete thermalization, the authors use a set of local or quasilocal charges with exponentially convergent tails. To quote the reference on string-charge duality [66] in this context does not seem very appropriate. I suggest to instead quote the review on quasilocal charges [JSTAT 064008 (2016)].

After my comments are carefully addressed I expect the paper to be suitable for publication in

Nature Communications.

Reviewer #2 (Remarks to the Author):

In this paper the authors study the out-of-equilibrium dynamics of a particular inhomogeneous system, namely one where at time zero two different semi-infinite spin chains are connected and let to evolve for two different choices of the initial state.

The paper makes three major claims:

- 1) Despite the fact that the time evolution is governed by a non-integrable hamiltonian, the large-time dynamics is not described by a thermal ensemble (in other words, the effect of the "junction" is somehow washed out in the large time limit, deep inside the light-cone).
- 2) The large-time (stationary) value of the energy current is proportional to a parameter which may be interpreted as a Kapitza boundary resistance.
- 3) The qualitative features of such a set-up are well reproduced by a model where the junction between chains is implemented by a chaotic hamiltonian acting on ℓ sites, as long as ℓ is sufficiently small.

The observation that no large-time thermalization occurs, even though the evolution hamiltonian is not integrable is not entirely new, as the authors rightfully point out. Theirs is rather a contribution to an already developing debate. However, their study is particularly detailed and the models they explore are distinct from cases previously considered. Also their detailed discussion of the distinct regions inside the lightcone is novel and would be worth investigating in more detail in future, perhaps through an extension of GHD techniques.

I believe the conclusions of the work to be new and original. The work provides a valuable contribution to a very active area of research and the set-up that they consider is one that could be of experimental interest, as they also point out.

However, in my view there is an element in the paper which requires further discussion and that is mainly the results presented in Fig. 4(b). I find it quite puzzling that the authors present such a plot without a more detailed discussion.

The disagreement between the results from hydrodynamic equations and the numerics is not just quantitative but also qualitative. The curves disagree in almost every possible way, in particular the presence of a local minimum and two local maxima in the black curve obtained from numerics is striking and plainly at odds with the hydro prediction.

I accept that the model described by the hydrodynamic picture and the model for which numerics is being performed are not identical, but if the connection to Kapitza resistance is to be taken seriously, then there should be at least a qualitative match.

The authors need to explain further why these curves are so different and ideally say something about the different monotonicity properties of the two curves (one set having a single maximum while the numerics shows two). Is there a physical interpretation for this?

Apart from this objection, I find that the paper is interesting and well written. There is a numerical part which of interest in itself as the implementation of MPS is non-trivial for this set-up. So up to clarifying the point above, I think the paper deserves to be published in NC and that is has potential to influence future research in the growing field of quench dynamics in integrable and non-integrable models.

Reply to reviewer 1

We thank the reviewer for her/his careful reading of our manuscript and for the helpful comments. We really appreciated the fact that she/he likes *the results and the reasoning of the paper* and that she/he thinks that *this is a very interesting and important problem*. We have made a number of changes following the reviewer's suggestions and in the following we reply to all her/his concerns:

- Q0. *the chaotic-impurity-region model assumes that the size of impurity-region ℓ does not grow with time. Under this assumption one easily obtains the Kapiza boundary resistance model, Eq. (7), and the resulting transport remains ballistic. But this model would then also predict that the current $J(n=0, t)$ would NOT decay as a function of t , right? How can this be reconciled with Fig. 6, where, say for $\gamma = \pi/4$, we notice decay of the central-bond current after times $t \sim 10 - 20$.*

I believe this issue will be hard to decide based on the available data which are probably state-of-the-art with the TEBD method. But at least a clear discussion of this seeming inconsistency should be in order. And perhaps commenting why authors believe that ℓ will not grow with t (perhaps for some small-enough growth one has no inconsistency with their no-go statement for thermalizing junction)?

- A0. We agree with the reviewer that, according to the chaotic-impurity-region model, the steady-state current should relax to a non-vanishing value. As the reviewer is correctly pointing out, our numerical data cannot discriminate between a situation in which the current is decaying to zero and another one in which the current saturates to a non-vanishing value. This problem is now addressed in more details in the text:

This plot [we here refer to current Fig. 5, previously called Fig. 6] poses a problem because we just pointed out that two regions with different temperatures build up close by, and we would thus expect a saturation of energy or magnetic current with time, even in the cases where $\Delta \neq 0$. It is thus an interesting question to assess the long-time limit of our model. Yet, our numerics witnesses an uninterrupted decaying trend. With the numerical methods at our disposal, we cannot make a conclusive statement about the longer-time behaviour of such current. However, in the next section we will argue that at the junction an homogeneous thermal state cannot be created: therefore, a finite current is generically expected to persist in the stationary state.

In particular, we say that, since no thermalization can occur exactly at the junction, it is possible that a finite current flows in the steady state.

Concerning the growth of the ℓ region, in the text we now comment on the Fig. 4a, where the numerics clearly shows that the region bridging the two thermal ones does not scale with time:

When inspecting in detail the local effective temperatures in Fig. 4a, we observe that two approximate plateaus appear to the left and to the right of the junction, corresponding to regions 3L and 3R. The two plateaus have different values and the interpolation between them takes place on a length scale of few sites and does not depend on time.

Additionally, when we introduce the model, we write:

The choice of a finite junction is motivated qualitatively by our numerical observations, although the model that we are here discussing requires ℓ to be much larger than the few sites considered numerically.

In our no-go theorem, we make no assumptions about the growth rate of the thermal region; we rather define “full thermalization” as the situation where a thermal region with a unique homogeneous temperature grows unbounded from the junction. We then show that this scenario is inconsistent with the dynamics within the two halves. While this argument suggests that full thermalization in this sense is not happening, it does not

provide any characterization of the stationary state. Therefore, although we do not expect the current to decay to zero, we have no prediction about how small the stationary value could be in practice.

Q1. *Page 2, left column: The author state that “Pure states are numerically more tractable than density matrices and allow us to explore longer time dynamics”. This is in contrast to the statement of a closely-related paper Nature Comm. 8, 16117 (2017) - which I believe should be cited - where the authors advocated density-matrix TEBD, at least in the linear response regime, as superior to pure-state TEBD in exactly the same context. The resolution of the dilemma is probably in the fact that the authors here are interested in very *specific* pure initial states, namely such as corresponding to ground (or extremal) eigenstates of the half-chains. For more generic pure (even though possibly separable) initial state, the situation would be very different. This is also clear from very moderate scaling of the entanglement entropies shown in Fig. (6). Also a comment on how different would be a regime of linear response, where one would make a slight step in β or magnetization across the junction, would be useful.*

A1. Simulating the dynamics of pure states in our setup is useful for two main reasons:

- they allow us to naturally assess the real-time entanglement properties (ill-defined in the case of mixed density matrices);
- they are computationally more treatable when working far from the linear-response regime (they allow us to reach longer times).

As correctly hinted by the reviewer, the second point is true only when evolving peculiar initial states, such as extremal spectral states. They do admit an efficient matrix-product-state representation (with small bond-link dimension) and generate a moderate amount of entanglement during their evolution. We remark that this statement is not true for arbitrary initial conditions and Hamiltonians.

In the reference mentioned by the reviewer (now cited in the article) the authors work in the linear-response regime starting from an almost fully mixed state with a very small amount of magnetization along the z direction (with opposite sign in the two halves). In this case, the mixed-state dynamics proves to be very effective. For this reason, in the early stages of our study, we performed several simulations in this regime and came to the conclusion that the slow time scale related to the appearance of regions 3L and 3R would be even slower. In fact, it was not numerically accessible. The study of a large energy/magnetization imbalance, far from the linear-response regime, is the best compromise that we found to deal with long time scales.

We added a few sentences in the introduction to clarify these points.

Q2. *Language is sometimes quite convoluted and consequently presentation is difficult to follow. For example, two sentences in the second paragraph, right column, page 2, starting with “Let us stress that...” were difficult to comprehend. I would suggest to replace the phrase “around the junction” with something more concise.*

A2. We thank the reviewer for her/his comment, we have reformulated an important part of the text and corrected several typos.

Q3. *In the argument about impossibility of a complete thermalization, the authors use a set of local or quasilocal charges with exponentially convergent tails. To quote the reference on string-charge duality [66] in this context does not seem very appropriate. I suggest to instead quote the review on quasilocal charges [JSTAT 064008 (2016)].*

A3. We have substituted the reference as suggested.

Reply to reviewer 2

We thank the reviewer for her/his helpful and constructive comments, and for the positive judgment, since she/he believes *that the paper is interesting and well written*. Moreover, after the clarification of a few points that are addressed here below, she/he thinks that *the paper deserves to be published in NC and that it has potential to influence future research*. We have taken into account her/his concerns regarding the disagreement between the theoretical model and the exact numerical data, and made a number of changes to our manuscript.

Q0. The reviewer's main concern is that *the disagreement between the results from hydrodynamic equations and the numerics is not just quantitative but also qualitative*. Referring to Fig. 4b, he continues:

The curves disagree in almost every possible way, in particular the presence of a local minimum and two local maxima in the black curve obtained from numerics is striking and plainly at odds with the hydro prediction. I accept that the model described by the hydrodynamic picture and the model for which numerics is being performed are not identical, but if the connection to Kapitza resistance is to be taken seriously, then there should be at least a qualitative match. The authors need to explain further why these curves are so different and ideally say something about the different monotonicity properties of the two curves (one set having a single maximum while the numerics shows two). Is there a physical interpretation for this?

A0. We have seriously taken into account the reviewer's suggestion, which motivated us to largely rewrite the **Results**. For instance, we now first present the numerical data, and only later we comment on their theoretical interpretation. This should avoid communicating the idea that numerics is presented to corroborate analytics, or the opposite. Rather, we want to present two different studies which display the same phenomenology: a discontinuity in the energy/temperature profile.

We agree with the reviewer on the fact that the exactly-solvable "toy model" is largely unsatisfactory in describing the numerical data. In the newer version we devote more space to comment on the disagreement between numerics and the analytical predictions of the chaotic model, and in particular we explicitly write:

The energy current is continuous by construction, but in this case the agreement with the numerics is poor.

To make the reader fully aware of the possible issues encountered when attempting to bridge numerics and analytics, we also added:

We conclude by mentioning that the energy current measured in our numerics still shows, at the longest accessible times, a significant decreasing trend. This is in contrast with the long-time predictions of the chaotic junction model and suggests a long relaxation time, as well as a large resistance. Understanding the microscopic origin and generality of this phenomenology are interesting questions that cannot be discussed with present numerical tools, and are thus left as an open issue for future investigations.

However, we have a physical interpretation of the disagreement between the two models. Indeed, numerical results distinguish between two kinds of regions inside the light cone:

- regions 2L and 2R, that are hydrodynamic-like;
- regions 3L and 3R, that are thermalizing.

The analytical model presents a stationary state that does not distinguish between the two kinds of regions. We believe that this is the main source of discrepancies. In the text we write:

In particular, in the analytical solution we do not observe two different regions inside the causal cone; but rather a unique one. We understand the distinction between regions 2 and

3 in our numerics as due to the different dynamics of fast and slow excitations. However, in the considered limit of a large chaotic junction, all quasiparticles undergo a large number of scattering events, irrespectively of their velocity..

In practice, we are suggesting that a large scattering region ℓ can renormalize even the fastest quasiparticles, thus washing out the differences between regions of type 2 and of type 3.

However, we point out that the theoretical model has only been introduced to reproduce the energy profiles and to show how a discontinuity in the energy/temperature profile is possible at $x/t = 0$, namely the fact that the temperature jumps, which originates what we call the Kapitza boundary resistance. In this way, we are able to present both one numerical analysis and one analytical model that show a similar phenomenology, corroborating our discussion in terms of a Kapitza boundary resistance.

We note that, while the toy model admits a simplified hydrodynamic treatment in terms of few mesoscopic parameters, an accurate description of the model we considered in our numerics would require a description of the scattering properties between quasiparticles of different kind which occurs at the junction. We expect this description to be model-dependent and so we do not attempt its analysis here inasmuch as we are more concerned with generic features of this protocol.

REVIEWERS' COMMENTS:

Reviewer #1 (Remarks to the Author):

I believe the authors have improved their manuscript in the revision process. The interpretations are substantially clearer now. It is also acknowledged that some issues cannot be settled based on the presently available data, such as for example, the asymptotic scaling of the current.

I recommend publication of the manuscript in NC in the present form.

Reviewer #2 (Remarks to the Author):

I thank the authors for considering my comments and acting on them. I think the changes that they have made improve readability and address my main criticism of the manuscript.

I am now happy with the way the manuscript is written and recommend it for publication.